# Will Buying Follow Others Ease Their Threat of Death? An Analysis of Consumer Data during the Period of COVID-19 in China

**DOI:** 10.3390/ijerph17093215

**Published:** 2020-05-06

**Authors:** Wei Song, Xiaotong Jin, Jian Gao, Taiyang Zhao

**Affiliations:** 1Business School, Jilin University, Changchun 130012, China; songweijlu@163.com (W.S.); jxtjlu@126.com (X.J.); 2School of Business and Administration, Zhejiang University of Finance and Economics, Hangzhou 310018, China; 3School of Philosophy and Sociology, Jilin University, Changchun 130012, China; taiyang5450@163.com

**Keywords:** threat of death, need to belong, materialism, perceived social support, informational conformity consumer behavior

## Abstract

How to overcome informational conformity consumer behavior when faced with threats of death is a social problem in response to COVID-19. This research is based on the terror management theory, the need to belong theory and the materialism theory. It uses a theoretical model to determine the relationships between threats of death and informational conformity consumer behavior. From 1453 samples collected during outbreak of COVID-19 in China, we used a structural equation model to test multiple research hypotheses. The result shows that threats of death are positively associated with a need to belong, materialism and informational conformity consumer behavior. The need to belong and materialism can play a mediating role between threats of death and information conformity consumption behavior, and perceived social support can play a moderating role between threats of death and information conformity consumption behavior.

## 1. Introduction

The spread of COVID-19 has proved to be disastrous for the whole world. The higher infection rate and the faster transmission speed present a life-threatening situation to people all over the world. This has not only caused great panic but has also changed our consumer behavior. If consumers hear that a product can suppress the virus, even if it is not confirmed, the product will be sold out almost immediately. Whether in the United States or China, face masks or disinfectants, Sanjing oral liquid today or Isatis root granules when SARS was first seen about 17 years ago, similar phenomena are continually emerging, and it seems that this problem is only growing. When consumers face the threat of death, their consumer behavior tends to be irrational, guided by the information of the group they belong to and resulting in conformity consumer behavior [1].

Conformity consumer behavior is one manifestation of social influence, resulting from the opposition of the other group members to an individual’s views [2]. It provides an adaptive shortcut that maximizes the likelihood of effective action with minimal expense to one’s cognitive resources [3]. Deutsch and Gerard (1955) identified two types of social influences: normative and informational [4]. Normative conformity behavior is the pressure to conform to the positive expectations of others. Information conformity behavior is the tendency to accept information from others as guidance when facing complicated information and wanting to simplify the decision-making process [5]. In the COVID-19 situation, the consumer choice is based on the information from others to help themselves mitigate the threat of death. Therefore, this phenomenon can be seen as an informational conformity consumer behavior. Previous studies found that informational conformity consumer behavior in severe epidemics will bring much harm, such as anxiety sensitivity [6], more crowded gathered, dramatic price change in a short time [7], impulsive consumer behavior [8], misallocation of resources and harm to the economy [9]. Therefore, controlling the informational conformity consumer behavior of consumers during an epidemic can not only relieve consumers’ anxiety, but also stabilize the economic and social order, making the plagues control more efficient to solve it.

This study primarily aimed to explore the influence mechanism between threat of death and information conformity consumer behavior and analyze the multi-mediating effect of the need to belong and materialism when they did the information conformity consumer behavior.

This study secondarily aimed to analyze the moderating effect of perceived social support and discuss the strategy to reduce the harm caused by information conformity consumer behavior in the COVID-19 situation.

## 2. Theoretical Background and Hypotheses

### 2.1. Terror Management Theory and Threat of Death 

The threat of death is one of the most critical risks possible to encounter [10]. When people foresee their own future death, negative emotions such as fear and anxiety will occur [11,12]. People will take the initiative to find positive emotions to deal with their negative emotions from the threat of death [13,14]. Therefore, the terror management theory says that death has a vast influence on the development of human society [11]. Moreover, all human motives are ultimately derived from a biologically based instinct for self-preservation [12]. The self-preservation system includes two psychological functions, one of which is the cultural worldview. This drives individuals to behave in a way that follows their culture and worldview [15]. When people think about their own death, they take sides with and defend their culture by increasing their support for worldview-consistent examples, decreasing support for worldview-threatening cases or both [16]. Another function is self-esteem, which motivates individuals to act in favor of their self-esteem [17,18] and choose high-priced products to show their value in society [19,20].

Previous studies have found that when consumers face the threat of death, to reduce fear and anxiety due to exposure, they will show differences in consumer behaviors. For instance, they may pay more attention to hedonic consumption behavior [21,22], brand consumption behavior, compulsive consumption behavior [23] and pro-social consumption behavior [24,25]. Therefore, threats of death often lead to irrational consumer behavior, and consumer behavior often involves decision-making under emotional anxiety.

### 2.2. Informational Conformity Consumer Behavior

Informational conformity consumer behavior is related to the external environment of the consumer [26]. It is based on the desire to form an accurate interpretation of reality and behave correctly. Previous studies have suggested that some factors lead to informational conformity consumer behavior. First is group status, the self-categorization theory posits that individuals categorize themselves at varying degrees of abstraction and use their social identities to reduce uncertainty when faced with prospective group conflict [27]. Clark et al. found that compared with the role of relaxed consumers, the status-seeking consumer would have more informational conformity consumer behavior [28]. Second is group consensus; the dynamic social impact theory (DSIT) posited that the group consensus is a higher-order cognitive state that emerges over time from local-level conformity within multiple person assemblages of varying sizes, functions, complexities and levels of interpersonal interaction [29]. It generates information in group interactions and influences members’ attitudes and behaviors [30], and social space could play a negative moderator role between them [31]. This result means that the larger the social space, the less likely it is that group consensus can influence informational conformity consumer behavior. If consumers enter a group, their attitudes and behaviors will inevitably be affected by the group, which becomes an essential condition for informational conformity consumer behavior.

Previous studies have shown a link between threats of death and informational conformity consumer behavior [1], and when a public crisis occurs, concern and trust between people become very important. Emotional communication with family, friends, and neighbors can bring psychological benefits to people, thereby reducing negative emotions caused by death information [32]. Moreover, the essential social motivations, such as self-protection, are the most easily communicated and potentially shared information among groups [33]. This social motivation forms the basis for consensus among groups and influences the behavior of group members. A severe public crisis like the COVID-19 outbreak gives all people a threat of death, which causes fear and anxiety. Communication between consumers and family, friends, and neighbors can reduce consumers’ fear and anxiety, and information about products related to self-protection, such as medicines and food, will become essential information to share among group members, thereby promoting conformity consumer behavior. Based on the above theory and logic analysis, the following hypothesis is proposed:

**Hypothesis 1** **(H1):***Threat of death is positively associated with informational conformity consumer behavior*.

### 2.3. Need to Belong

The need to belong is innate for human beings and motivates people to form and maintain social relationships [34]. It operates in a wide variety of settings and guides emotional, cognitive and behavioral responses. The theory states that everyone needs a minimum number of caring, interpersonal relationships that are stable, ongoing, frequent, mutual and primarily free from conflict. The key to the need to belong is a stable social relationship, and it satisfies two criteria: First, there is a need for frequent, affectively pleasant interactions with a few other people, and, second, these interactions must take place in the context of a temporally stable and enduring framework of affective concern for each other’s welfare. Previous studies found that when the need to belong is satisfied, three patterns of positive response outcomes are gained. The first outcome is a pattern of positive emotional reactions that are generally characterized as joyful. The second outcome is a thinking pattern that promotes a pervasive concern to maintain caring, interpersonal relationships. The third outcome is a pattern of goal-directed activity designed to satisfy the need to belong continually. Failure to meet this need may result in a variety of adverse effects on adjustment and well-being, such as illness and depression [34]. Previous studies have found that personal self-interest [35] and social exclusion [36] could negatively influence the need to belong. Comfortable environment and food [37], parasocial interaction [38], nostalgia proneness [39] and loneliness [40] could positively influence the need to belong. Some factors could be influenced by the need to belong, such as belief in God [41,42], happiness [43] and social interactions. Therefore, the need to belong takes place in situations of social interaction. It affects consumers’ emotional and behavioral choices through social interaction.

In the relationship between threats of death and the need to belong, previous studies have found that when faced with threats of death. The consumer seeks out the need to belong to reduce their fear and anxiety. For example, Jones, Parker-Raley and Barczyk’s (2011) study about adolescent cancer survivors posit that the threats of death can cause adolescent cancer survivors to feel isolated, so the need to belong can alleviate the anxiety and fear caused by the threat of death [44]. This conclusion shows that when consumers face threats of death, such as COVID-19, they can satisfy their need to belong in the group by improving social interaction to overcome anxiety and fear, which means that the threats of death can increase consumers’ need to belong. The following hypothesis is proposed:

**Hypothesis 2** **(H2):**
*The threat of death is positively associated with the need to belong.*


The consumer facing threats of death would increase the need to belong [44]. Baumeister and Leary (1995) argued that the increase in the need to belong should lead to an increased cognitive focus on relationships and social connections [34]. Its inevitability will improve individual social interactions and increase the opportunity to perceive opinion consensus on essential issues [45]. The consensus on essential issues has become an important prerequisite for informational conformity consumer behavior. Regarding COVID-19, the consumer faces the threat of death by the virus and feels the need to belong to a group to lessen the anxiety and fear. Due to the social interaction within the group, it not only satisfies the need to belong, but also makes it easier to obtain consensus within the group and then generates informational conformity consumer behavior. Thus, this study proposed the following hypothesis:

**Hypothesis 3** **(H3):**
*Need to belong can play a mediating role between threat of death and informational conformity consumer behavior.*


### 2.4. Materialism

Materialism is a consumption-orientation and is defined as the importance a consumer attaches to worldly possessions. At the highest levels of materialism, consumers place material property at the core of their lives and use it as a measure of their satisfaction [46]. Previous studies have found that materialism is universal in cross-cultural comparisons and is not directly related to affluence [47]. There are two different research frameworks for the emergence of materialism. One is the developmental model, which holds that materialism is formed by the interaction of daily event cycles, developmental tasks, cultural influence and family environment [48]. The other is the reinforcement model, which posits that consumers’ differences in personal qualities make them generate psychological threats in daily events, thus leading to psychological discomfort. In order to reduce this discomfort, consumers adopt materialistic behaviors to improve their ability to change [48].

The terror management theory found that when consumers faced the threat of death, their cultural worldview was activated to drive their behavior following their cultural worldview of material pursuits [15]. The consumers have become materialistic and greedy, more willing to buy luxury goods and more inclined to hedonic consumption behavior [21,22]. In the context of the spread of infectious diseases such as COVID-19, consumers face a severe threat of death, which has also led to the activation of their cultural worldview of material pursuits, resulting in a strong materialist tendency. Therefore, the threat of death is positively associated with materialism, this is, materialism represented by money and wealth often becomes a tool for consumers to defend against threats of death. The following hypothesis is proposed:

**Hypothesis 4** **(H4):***Threat of death is positively associated with materialism*.

The human awareness of death affects materialism and increases people’s pursuit of wealth and culturally desired commodities to protect the existential anxieties from threats of death [21,49]. Consumers’ psychological conditions will inevitably lead to matching behaviors and manifest as changes in consumer behavior. Infectious diseases, such as COVID-19, lead to consumers facing the threat of death, so in order to reduce the resulting external anxiety, they may use the goods related to the reduction of the pandemic as a vital sign of improving their security. After the group provides relevant information, it will generate collective consciousness, which will lead to informational conformity consumer behavior. This means that materialism could have an indirect effect between threats of death and informational conformity consumer behavior. The following hypothesis is proposed:

**Hypothesis 5** **(H5):**
*Materialism can play a mediating role between threat of death and informational conformity consumer behavior.*


Previous studies have found that threats of death could affect materialism [21,49]. However, the indirect mechanism was not precise. Some studies suggested that in a social group, the relationship between threats of death and materialism and the need to belong could play vital roles. On one hand, the threat of death may influence the need to belong [44]. On the other hand, Baumeister and Leavy (1995) suggested that materialism is an essential part of human culture [34]. The purpose of individuals needing to belong is to satisfy the psychological and economic needs of living together in the group. Therefore, people with a strong need to belong may be predisposed materialism [50]. Based on the above theoretical reasoning, we can assume that when individuals face the threat of death (e.g., COVID-19), their materialism will increase, and the need to belong will play a mediating role between threat of death and materialism. The following hypothesis is proposed:

**Hypothesis 6** **(H6):**
*Need to belong can play a mediating role between threats of death and materialism.*


### 2.5. Perceived Social Support

Perceived social support operates in part as a cognitive personality construct [51]. It comes from friends, family and significant others [52]. Increasing individuals perceived social support of individuals was increased, they can help them overcome negative emotions (such as fear and anxiety) caused by disasters and injuries [53,54,55]. Perceived social support can improve their self-efficacy [54], positively relate to social goal pursuit [56] and influence life satisfaction [57,58] and well-being [59,60]. Individuals with higher perceived social support are more likely to engage in group activities [61] and have a more robust group identification [62].

Perceived social support is a psychology placebo for individuals facing negative emotions and overcoming negative emotions such as depression, anxiety and fear. When consumers face the threat of death, they seek social support from the group to secure attachment to help reduce anxiety and fear by the threat of death [63]. If their perceived social support were increased, then their identity to the group will be strengthened. After they understand the information consensus within the group, they will reduce the complexity of their behavioral decisions and conduct conformity consumption behavior.

**Hypothesis 7** **(H7):**
*Perceived social support can play a moderating role between threats of death and information conformity consumer behavior.*


Based on the above discussion, Figure 1 shows the proposed conceptual model of this study.

## 3. Methods

This section describes the data collection process for this study (Section 3.1), the establishment of measures for threat of death, need to belong, materialism, perceived social support and information conformity consumer behavior for it (Section 3.2), as well as the analytical approach employed (Section 3.3).

### 3.1. Survey Participants

Since many cities in China were subject to stay home orders due to the need for epidemic prevention at the time of the survey, this research commissioned Wenjuanxing Inc. (WJX), a professional market research company, to conduct this survey. We used the random sampling from the database of WJX samples and sent the questionnaire by email. The survey was conducted from 1 February to 20 February 2020. The participants for this study were 1499 individuals aged 18 to 79 who participated in the consumer behavior data survey during the COVID-19 outbreak in China. The data came from the participants in all 31 provinces of China and there were 1453 valid questionnaires. The net response rate was 93.86%. The sample included 645 female participants (44.4%) and 808 male participants (55.6%) and had a large share in income is 3000 RMB to 6000 RMB per month (47.40%) and a significantly higher level education (73.91% of participants had completed a college or university program). The survey also contains information on individual, family, area characteristics and the attitude of the government towards the virus (e.g., alertness level), which we can link to the attitudes and behavior of individuals.

### 3.2. Measure

SPSS 25.0 (SPSS Inc., Chicago, IL, USA) and AMOS 24.0 (SPSS Inc., Chicago, IL, USA) used scale structure, reliability analysis and convergent validity, and the measurement scales employed in this research were developed and validated in a past study. For information conformity consumer behavior, we used a three-item scale adapted from Bearden et al.’s (1989) study [64]. The Cronbach’s α was 0.744, the average variance extracted (AVE) was 0.504, and the composite reliability (C.R.) was 0.745. For the need to belong, we used a three-item scale adapted from Leary et al.’s (2013) study and in this study [65]. The Cronbach’s α was 0.686, the AVE was 0.424 and the C.R. was 0.675. Materialism was measured based on Richins and Dawson’s (1992) study and used an eight-item scale [66]. The Cronbach’s α was 0.860, the AVE was 0.439 and the C.R. was 0.861. Furthermore, the perceived social support was measured using a six-item scale from Canty-Mitchell and Zimet’s (2000) study [67]. The Cronbach’s α was 0.839, the AVE was 0.465 and the C.R. was 0.839 (See Table A1).

For the threat of death, according to Tobacky (1983)’s threat-of-death scale and the situation of this study [68], we designed a three-item scale to measure the construct. The content of this scale was that “What do you think is the risk of contracting COVID-19”, “How scared you are of a COVID-19” and “How much isolated are you?” In order to verify the reliability of the threat-of-death scale, this study randomly divided the research participants into two groups (One group has 726, and the other group has 727) and tested them with exploratory factor analysis (EFA) and confirmatory factor analysis (CFA), respectively. The results found that the EFA group’ s Cronbach’s α was 0.758, the AVE was 0.533 and the C.R. was 0.769. The CFA group’s Cronbach’s α was 0.705, the AVE was 0.482 and the C.R. was 0.724. This proves that the reliability and validity of the threat-of-death scale are relatively stable and can be used for subsequent analysis. The Cronbach’s α of the total sample was 0.732, the AVE was 0.504 and the C.R. was 0.745.

All of the above measurements used a 5-point Likert scale, and except for the need to belong (Cronbach’s α = 0.682, near 0.700), the Cronbach’s α of other constructs all exceeded 0.700. Except for the need to belong (AVE = 0.424, near 0.500), materialism (AVE = 0.437, near 0.500) and perceived social support (AVE = 0.465, near 0.500), the AVE of other constructs all exceeded 0.500, exhibiting sufficient reliability and convergent validity (See Table 1).

We used the correlation coefficient matrix of all constructs to examine the discriminant validity. The intercorrelations among the variables are reported in Table 2. The results indicated that the correlation relationships of all variables were all significant. Meanwhile, the correlation coefficients of related variables are smaller than the square root of the corresponding variable AVE on the diagonal, this proves that the constructs have good discrimination validity.

### 3.3. Analytical Process

To examine the measurement properties of the threat of death, need to belong, materialism, perceived social support and information conformity consumer behavior, the confirmatory factor analyses (CFA) were conducted. The CFA aims to identify a small number of factors by a set of observed indicators. In other words, CFA examines the dimensionality of a theoretical construct by analyzing interrelationships among stochastic indicators of this construct. To perform CFA for the threat of death, need to belong, materialism, perceived social support and information conformity consumer behavior, the software AMOS 24.0 was used.

To explore the relationships between these latent constructs and to control for certain relevant observed variables, structural models were estimated. For this purpose, a maximum likelihood estimator was used, being the most widely used fitting function for general structural equation models. A variety of fit indices have been developed, and for this study, the recommendations of Kline (2016) and Hooper, Coughlan and Mullen (2008) were followed, with the chi-squared test reported [69,70]. In addition, RMSEA, as a measure of the discrepancy per degree of freedom that was strongly recommended for evaluating model fit [71], was used. If the approximation is reasonable, RMSEA is small, with a value of 0.08 or less, suggesting a close fit [72].

## 4. Results

### 4.1. Measurement Model

As described in Section 3.3, a confirmatory factor analysis of the construct scales was performed, including the five first-order factors (threat of death, need to belong, materialism, perceived social support and information conformity consumer behavior), each represented by three or eight scale items. The results showed a CFI value of 0.930 and a RMSEA value of 0.059. The chi-square value was 682.245 and the chi-square/df value was 6.038. The model fit of this measurement model was satisfactory

### 4.2. Direct Effect Test

This study used the structural equation method (SEM) to test the hypotheses (see Table 3). The results showed that the threat of death was positively significantly affected the information conformity consumer behavior. The coefficient of the path was 0.396, the T-value was 8.381 and *P* < 0.001, which means that the threat of death improves the production of information conformity consumer behavior. Thus, H1 was supported. Meanwhile, the threat of death was positively significantly affected the need to belong (β = 0.411, S.E = 0.050, T-value = 8.262, *p* < 0.001) and materialism (β = 0.125, S.E = 0.029, T-value = 4.369, *p* < 0.001). This means that when consumers face a higher threat of death, they feel a greater need to belong and exhibit more materialism. Thus, H2 and H4 were supported.

### 4.3. Indirect Effect Test

This study used the bootstrapping method to test the indirect effect. With a bootstrapping sample conducted 2000 times, the results showed that: in the relationship between threat of death and information conformity consumer behavior, the total effect is significant, the Z value is 8.559 (*p* < 0.001), the value of the bias-corrected 95% CI lower is 0.403, the upper is 0.628, and it did not include zero. Moreover, the value of percentile 95% CI lower is 0.401, and the upper is 0.626, it did not include zero too. The result again proved the H1 was supported. Meanwhile, the indirect effect is significant, the Z value is 5.450 (*p* < 0.001), the value of the bias-corrected 95% CI lower is 0.077, and the upper is 0.157, it did not include zero. Furthermore, the value of percentile 95% CI lower is 0.073, and the upper is 0.153, it did not include zero too. Moreover, the direct effect is significant too, the Z value is 7.200 (*p* < 0.001), the value of the bias-corrected 95% CI lower is 0.300, and the upper is 0.509, it did not include zero. Moreover, the value of percentile 95% CI lower is 0.301, and the upper is 0.515, it did not include zero too. The result means that there was a significantly partial indirect effect between the threat of death and information conformity consumer behavior, at the same time, the threat of death could positively significantly affect the need to belong (β = 0.411, S.E = 0.050, T-value = 8.262, *p* < 0.001) and the need to belong could positively significantly affect the information conformity consumer behavior (β = 0.119, S.E = 0.036, T-value = 3.294, *p* < 0.001). Hence, the need to belong can play a significantly mediate role between the threat of death and information conformity consumer behavior. The H3 was supported.

Meanwhile, the threat of death was positively significantly affecting the materialism (β = 0.125, S.E = 0.029, T-value = 4.369, *p* < 0.001), and the materialism was positively significantly affecting the information conformity consumer behavior (β = 0.230, S.E = 0.053, T-value = 4.341, *p* < 0.001). Hence, materialism can play a significantly mediate effect between the threat of death and information conformity consumer behavior. The H5 was supported.

In the relationship between threat of death and materialism, the results showed that the total effect is significant, the Z value is 6.366 (*p* < 0.001), the value of the bias-corrected 95% CI lower is 0.188, the upper is 0.354, and it did not include zero. Moreover, the value of percentile 95% CI lower is 0.184, the upper is 0.342, and it also did not include zero. The result again proved the H1 was supported. Meanwhile, the indirect effect is significant, the Z value is 6.182 (*p* < 0.001), the value of the bias-corrected 95% CI lower is 0.096, the upper is 0.184, and it did not include zero. Moreover, the value of percentile 95% CI lower is 0.096, the upper is 0.184, and it did not include zero. In addition, the direct effect is significant, the Z value is 3.676 (*p* < 0.01), the value of the bias-corrected 95% CI lower is 0.064, the upper is 0.197, and it did not include zero. Furthermore, the value of percentile 95% CI lower is 0.060, the upper is 0.190, and it did not include zero. These results mean that there was a significantly partial indirect effect between the threat of death and information conformity consumer behavior, and that need to belong can play a mediating role between the threat of death and materialism. Thus, H6 was supported (See Table 4).

### 4.4. Moderating Effect Test

This study uses hierarchical regression to analyze the moderating effects. If the interaction item is significant, it means that the moderating effect was supported. Before the final calculation, all data were centralized to reduce the possible multicollinearity of the data. In the relationship between the threat of death and the information conformity consumer behavior, the M3 result showed that the value of DT × PSS is 0.467, the T-value is 3.343, *p* < 0.01 (R^2^ = 0.123, Adjust R^2^ = 0.121, ΔR^2^ = 0.007, F-change = 11.176), which means that the perceived social support can play a significant moderating role between the threat of death and information conformity consumer behavior. Thus, H7 was supported (See Table 5).

## 5. Discussion

The current study investigated the relationships among the threats of death, the need to belong, materialism, perceived social support and information conformity consumer behavior in the context of COVID-19. The results indicated that the threats of death are positively associated with the need to belong, materialism and informational conformity consumer behavior. The need to belong and materialism can mediate between threats of death and information conformity consumption behavior, and perceived social support can play a moderating role between threats of death and information conformity consumption behavior.

### 5.1. Theoretical Contribution

Our findings further information conformity consumer behavior in the context of COVID-19 in two ways. First, we have proposed a new theoretical explanation for the impact of the threats of death faced by consumers on information conformity consumer behavior in the context of COVID-19. Second, we have analyzed the multiple mediating effects between threats of death and information conformity consumer behavior and the boundary conditions in which perceived social support plays a role.

The result of the relationship between threats of death and information conformity consumer behavior verifies the terror management theory [11,12,19]. The result from the testing of H3, which predicted that the need to belong mediates the relationship between threats of death and information conformity consumer behavior, aligns with the findings of prior studies on the need to belong theory [34,35]. When consumers face the threat of death brought about by the spread of COVID-19, they are expecting to meet their belonging needs through group activities, and then overcome their fear and anxiety caused by the threat of death, and the identification and conformity of the information within the group is an important step to obtain the approval to the group members. Thereby, this is a change in the behavior of consumers and their consumer behavior induces informational conformity consumer behavior.

The result from the testing of H4 and H5 indicated that when individuals face threats of death, in order to overcome the anxiety and fear brought about by the threat of death, while seeking the need to belong for group belonging, they must ultimately relieve their negative emotions through hoard material resources and communicate more with group member [48,73]. This phenomenon is just like a Chinese proverb. “There is food in the home, so do not panic.”

The result from the testing of H7, which predicted that perceived social support moderates the relationship between threats of death and information conformity consumer behavior, verifies the theory of perceived social support. When the perceived social support of individuals was increased, they were better able to overcome negative emotions caused by disasters and injuries [53,54,55]. Then if consumers get social support from the group, it will in turn give the group stronger identification. After they understand the information consensus within the group, they will reduce the complexity of their behavioral decisions and use conformity consumption behavior.

### 5.2. Managerial Implications

The current study provides several managerial implications for the practice of public administration in the context of the epidemic.

First, information about the epidemic should be published in a timely and transparent manner. In addition, the government should give correct guidance about the products related to the epidemic in a timely manner, to help consumers understand the correct anti-epidemic products and life-related knowledge, and to avoid conformity consumption behavior caused by misinformation. In the survey, we found that many consumers were misled by their social groups, and they hoarded many products that did not have any effect against COVID-19. Consumers did not rationally evaluate the effects of these products but followed others via conformity consumption behavior. This not only misleads consumers about epidemic prevention, but also causes a waste of resources. Therefore, at the beginning of an epidemic, it is vital to give consumers correct information and proper guidance.

Second, the individual should be separated from the group. Individuals in the group will feel the social support given by the group, but will also get the misleading information, which will not benefit their epidemic prevention work. A large number of conformity consumption behaviors bring about a shortage of related products, especially in the process of epidemic prevention. A large number of people gathering will also put people at more severe risk of infection. The government can adopt centralized management, on-demand distribution models, use of logistics delivery and other methods to of delivery, to ease consumer anxiety and reduce the risk of infection.

Third, the government should intervene in the distribution of anti-epidemic products, which cannot be completely handed over to the market mechanism. In particular, the government needs to use laws and regulations to crack down on hoarding, illegal trade, and maliciously driven up prices to maintain market order. The market mechanism has the characteristics of blindness and spontaneity. When the epidemic causes conformity consumer behavior, a large number of product demands will cause prices to rise sharply, and some unscrupulous merchants will use vending to conduct vicious competition. If not controlled in time, it will cause serious damage to the market order and affect consumers’ confidence. Therefore, it is necessary to use legal tools to crack down on merchants using illegal methods.

Finally, when faced with the threat of death from the epidemic, people’s lives are the most important thing. When the government formulates any laws and economic management policies, it should take respect for people’s lives into account as a prerequisite, coordinate the contradiction between life and economic interests, and when life and economic interests conflict, resolutely choose the policy most helpful to people’s lives.

### 5.3. Limitations and Direction for Further Research

Despite the contributions of this study, there are limitations, some of which can be the basis for future research. First, our research used the survey method to verify our research hypotheses. Using an experimental method could ensure the external validity of the research to a certain extent, especially in the particular research context of COVID-19. However, it may inevitably be affected by consumer characteristics and differences in threats of death in different regions. Future research can use experimental methods further to verify the relationships between the variables in this study. Second, the sample of this study is from China. The collective consciousness of Chinese people is different from that in other countries, which reduces external validity. Future research may be conducted in different countries to further verify the hypotheses of this study.

## 6. Conclusions

This present study highlights the association between death threat and informational conformity consumer behavior in the situation of COVID-19. In addition, it explores the indirect effects of need to belong and materialism, and the moderating role of perceived social support. The government should use public management power to reduce the people’s anxieties caused by death threats, alleviate the shortage of materials due to informational conformity consumer behavior, and block the possibility of the spread of the virus widely.

## Figures and Tables

**Figure 1 ijerph-17-03215-f001:**
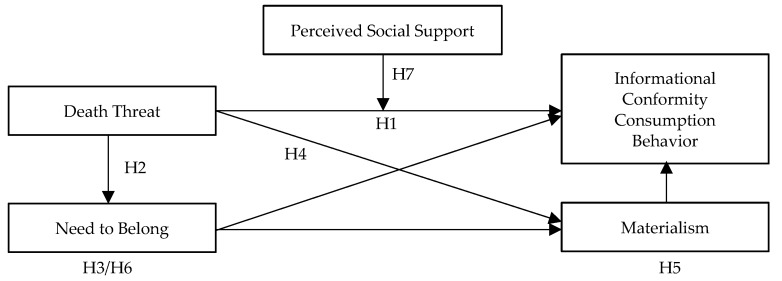
Conceptual model.

**Table 1 ijerph-17-03215-t001:** Reliability and Validity Analysis (*N* = 1453).

Variable Name	Item	Unstandardized Factor Loading	Standard Error	T-Value	*p*	Standardized Factor Loading	C.R.	AVE	Cronbach’s α
Death Threat (DT)	DT1	1				0.759	0.745	0.504	0.732
	DT2	1.057	0.060	17.477	***	0.825			
	DT3	0.472	0.029	16.391	***	0.504			
Informational Conformity Consumption Behavior (ICCB)	ICCB1	1				0.623	0.750	0.502	0.744
	ICCB2	1.170	0.064	18.138	***	0.789			
	ICCB3	1.139	0.061	18.653	***	0.704			
Materialism (MA)	MA1	1				0.696	0.861	0.439	0.860
	MA2	0.819	0.037	22.173	***	0.639			
	MA3	0.989	0.038	25.818	***	0.745			
	MA4	0.743	0.037	20.321	***	0.607			
	MA5	0.835	0.038	22.099	***	0.644			
	MA6	1.068	0.043	24.853	***	0.761			
	MA7	0.846	0.038	22.531	***	0.676			
	MA8	0.630	0.036	17.325	***	0.498			
Need to Belong (NB)	NB1	1				0.409	0.675	0.424	0.686
	NB2	2.163	0.172	12.554	***	0.780			
	NB3	2.153	0.172	12.537	***	0.781			
Perceived Social Support (PSS)	PSS1	1				0.683	0.839	0.465	0.839
	PSS2	1.080	0.046	23.706	***	0.711			
	PSS3	0.974	0.049	19.803	***	0.641			
	PSS4	0.956	0.048	19.942	***	0.649			
	PSS5	1.082	0.046	23.571	***	0.730			
	PSS6	1.024	0.047	21.783	***	0.674			
Model Fit	Chi-square = 682.245, df =113, Chi-square /df = 6.038, CFI = 0.930, TLI = 0.915, NFI = 0.917, RMSEA = 0.059.								

Note: *** *p* < 0.001.

**Table 2 ijerph-17-03215-t002:** Mean, standard deviation and correlation coefficient matrix (*N* = 1453).

	Mean	Standard Deviation	1	2	3	4	5
DT	3.9202	1.2337	0.710				
MA	2.9783	0.7639	0.224 ***	0.663			
NB	3.1936	0.7742	0.236 ***	0.431 ***	0.651		
ICCB	2.8674	0.8877	0.324 ***	0.288 ***	0.252 ***	0.708	
PSS	3.6432	0.7121	0.063 *	−0.059 *	0.138 ***	0.124 ***	0.682

Note: * *p* < 0.05; *** *p* < 0.001. The off-diagonal numbers are the correlations. AVE square roots are bolded on the diagonal.

**Table 3 ijerph-17-03215-t003:** The test of direct effect (*N* = 1453).

Path	Estimate	S.E.	T-Value	*p*
DT—>NB	0.411	0.050	8.262	***
DT—>MA	0.125	0.029	4.369	***
NB—>MA	0.330	0.027	12.083	***
DT—>ICCB	0.396	0.047	8.381	***
MA—>ICCB	0.230	0.053	4.341	***
NB—>ICCB	0.119	0.036	3.294	***

Note: *** *p* < 0 001.

**Table 4 ijerph-17-03215-t004:** The test of indirect effects (Bootstrapping times = 2000, *N* = 1453).

Path	Effect	Point Estimate	Coefficient			Bootstrapping			
						Bias-Corrected 95% CI		Percentile 95% CI	
			SE	Z	*p*	Lower	Upper	Lower	Upper
DT—>ICCB	Total effect	0.505	0.059	8.559	***	0.403	0.628	0.401	0.626
	Indirect effect	0.109	0.020	5.450	***	0.077	0.157	0.073	0.153
	Direct effect	0.396	0.055	7.200	***	0.300	0.509	0.301	0.515
DT—>MA	Total effect	0.261	0.041	6.366	***	0.188	0.354	0.184	0.342
	Indirect effect	0.136	0.022	6.182	***	0.096	0.184	0.096	0.184
	Direct effect	0.125	0.034	3.676	**	0.064	0.197	0.060	0.190
NB—>ICCB	Total effect	0.195	0.035	5.571	***	0.129	0.271	0.124	0.266
	Indirect effect	0.076	0.021	3.619	***	0.039	0.121	0.038	0.118
	Direct effect	0.119	0.042	2.833	**	0.038	0.207	0.035	0.205

Note: ** *p* < 0.01; *** *p* < 0.001.

**Table 5 ijerph-17-03215-t005:** The test of moderating effect.

Variable Name	ICCB					
	M1		M2		M3	
	β	T	β	T	β	T
DT	0.324 ***	13.066	0.318 ***	12.848	−0.067	−0.569
PSS			0.104 ***	4.211	−0.125	−1.715
DT × PSS					0.467 **	3.343
R^2^	0.105		0.116		0.123	
Adjust R^2^	0.105		0.115		0.121	
ΔR^2^	0.105		0.011		0.007	
F charge	170.716		17.737		11.176	

Note: ** *p* < 0.01; *** *p* < 0.001.

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
