# Peer review of "Will Buying Follow Others Ease Their Threat of Death? An Analysis of Consumer Data during the Period of COVID-19 in China"

_ijerph, 2020, doi:10.3390/ijerph17093215_

Round 1
Reviewer 1 Report
In the current context of the pandemic with COVID-19, a scientific article on consumer behavior facing the dead threats is welcome and will certainly provide a solid basis for any future research.
This research analyzes the 7 hypotheses proposed and reaches relevant results. However, there are some recommendations:
- Figure 1 Conceptual model shows the relationships between the 7 hypotheses proposed for this research. Perhaps there is a variant for H1 to be better represented visually in this scheme.
- At paragraph 3.1 Survey Participants, the period for conducting the online research and the sampling method used should be mentioned more clearly.
- I think there is a misunderstanding about the age of the people in this survey. The age group of 13-18 years, minors, are really representative? Can have responsible behavior regarding the purchase of some products related to the epidemic in time, taking into account that they don’t have an income from work, or is it a particular situation in China?
- In order to have a better consistency and a high scientific level, I suggest mentioning some managerial implications at economic level in order to support the groups of consumers with risk degree in the case of this extremely severe pandemic with COVID-19.
Author Response
Dear reviewer
Thanks for your review, your suggestion has a very great guiding value for our work. Follow your advice for our article, we have made major changes to the article and will report the changes to you.
Question1
Figure 1 Conceptual model shows the relationships between the 7 hypotheses proposed for this research. Perhaps there is a variant for H1 to be better represented visually in this scheme.
Reply:
Thanks to the expert’s suggestion, we have redrawn Figure 1 in order to better express the research framework of this study. We have added new pictures in the attachment.
Suggestion2
At paragraph 3.1 Survey Participants, the period for conducting the online research and the sampling method used should be mentioned more clearly.
Reply:
Thanks to the expert’s suggestion. The original description of the survey participants and the sampling method are indeed not clear enough. We have made a more detailed description in the revision. We hope that our revision will make the readers more understand our research process.
Before:
3.1 Survey Participants
Respondents for this research were recruited from WJX, which is the most prominent Chinese online data collection platform. The participants for this study were 1548 individuals aged 13 to 79 who participated in the consumer behavior data survey during the period of COVID-2019 in China. The data came from the participants in all 31 provinces of China undergoing COVID-2019 and had 1502 valid questionnaires. The net response rate was 97.03%. The sample included 664 female participants (44.2%) and 838 male participants (55.8%) and had a large share in income is 3000RMB to 6000RMB per month (45.87%) and a significantly higher level education (72.24% of participants had completed a college or university program). The survey also contains information on the individual, family, area characteristics, and the attitude of the government towards the virus, which we can link to the attitudes and behavior of individuals to the COVID-2019 in China.
After:
3.1 Survey Participants
Since many cities in China were subject to stay home orders due to the need for epidemic prevention at the time of the survey, this research commissioned WJX, a professional market research company, to conduct this survey. We used the random sampling from the database of WJX samples and sent questionnaire by email. The survey was conducted from February 1 to February 20, 2020. The participants for this study were1499 individuals aged 18 to 79 who participated in the consumer behavior data survey during the COVID-19 outbreak in China. The data came from the participants in all 31 provinces of China and there were 1453 valid questionnaires. The net response rate was 93.86%. The sample included 645 female participants (44.4%) and 808 male participants (55.6%) and had a large share in income is 3000RMB to 6000RMB per month (47.40%) and a significantly higher level education (73.91% of participants had completed a college or university program). The survey also contains information on individual, family, area characteristics, and the attitude of the government towards the virus(e.g. alertness level), which we can link to the attitudes and behavior of individuals.
Suggestion3:
I think there is a misunderstanding about the age of the people in this survey. The age group of 13-18 years, minors, are really representative? Can have responsible behavior regarding the purchase of some products related to the epidemic in time, taking into account that they don’t have an income from work, or is it a particular situation in China?
Reply:
Thanks to the expert’s suggestion. We agree with you, so in the sample we deleted all samples under the age of 18, and calculated and re-analyzed the data. The results also support the hypothesis proposed in this study.
Suggestion4:
In order to have a better consistency and a high scientific level, I suggest mentioning some managerial implications at economic level in order to support the groups of consumers with risk degree in the case of this extremely severe pandemic with COVID-19.
Reply:
Thanks to the expert’s suggestion. Actually, it is very necessary to put forward specific management suggestions. As the corresponding author, I am currently conducting research on visiting scholars in the United States. By comparing the management policies of the United States and China at COVID-19, we believe that all policies should be based on protecting people ’s lives. We suggest that the government's power should be fully used to maintain market order, rather than letting the market adjust itself spontaneously. Based on this principle, we further put forward the following suggestions.
Before:
5.2 Managerial Implications
The current study provides several managerial implications for the practice of public administration in the context of the epidemic.
First, Information about the epidemic situation should be published in a timely and transparent manner, and residents should be given proper guidance. The government should give correct guidance to the products related to the epidemic in time, to help consumers understand the correct anti-epidemic products and life-related knowledge, and to avoid consumers' following the conformity consumption behavior caused by misinformation in the group. In the survey, we found that many consumers were misled by their social groups, and they hoarded many products that did not have any effect against COVID-2019. Consumers did not rationally evaluate the effects of these products but followed others and conduct the conformity consumption behavior, which not only misleads consumers on their epidemic prevention work but also causes a waste of resources. So at the beginning of the epidemic, it is vital to give consumers the correct information and the right guidance.
Second, separate the individual from the group. Individuals in the group will feel the social support given by the group, but will also get the misleading information in the group more irrationally, which will harm their epidemic prevention work. A large number of conformity consumption behaviors bring about a shortage of related products, especially in the process of epidemic prevention. A large number of people gathering will also put people at a more severe risk of infection. The government can adopt centralized management, on-demand distribution model, using logistics delivery, and other methods to deliver, to ease consumer anxiety and reduce the risk of infection of ordinary people.
After:
5.2 Managerial Implications
The current study provides several managerial implications for the practice of public administration in the context of the epidemic.
First, information about the epidemic should be published in a timely and transparent manner. In addition, the government should give correct guidance about the products related to the epidemic in a timely manner, to help consumers understand the correct anti-epidemic products and life-related knowledge, and to avoid conformity consumption behavior caused by misinformation. In the survey, we found that many consumers were misled by their social groups, and they hoarded many products that did not have any effect against COVID-19. Consumers did not rationally evaluate the effects of these products but followed others via conformity consumption behavior. This not only misleads consumers about epidemic prevention, but also causes a waste of resources. Therefore, at the beginning of an epidemic, it is vital to give consumers correct information and proper guidance.
Second, the individual should be separated from the group. Individuals in the group will feel the social support given by the group, but will also get the misleading information, which will their epidemic prevention work. A large number of conformity consumption behaviors bring about a shortage of related products, especially in the process of epidemic prevention. A large number of people gathering will also put people at more severe risk of infection. The government can adopt centralized management, on-demand distribution models, use of logistics delivery, and other methods to of delivery, to ease consumer anxiety and reduce the risk of infection.
Third, the government should intervene in the distribution of anti-epidemic products, which cannot be completely handed over to the market mechanism. In particular, the government need to use laws and regulations to crack down on hoarding, illegal trade, and maliciously driven up prices to maintain market order. The market mechanism has the characteristics of blindness and spontaneity. When the epidemic causes conformity consumer behavior, a large number of product demands will cause prices to rise sharply, and some unscrupulous merchants will use vending to conduct vicious competition. If not controlled in time, it will cause serious damage to the market order and affect consumers' confidence. Therefore, it is necessary to use legal tools to crack down on merchants using illegal methods.
Finally, when faced with the threat of death from the epidemic, people’s lives are the most important thing. When the government formulates any laws and economic management policies, it should take respect for people's lives into account as a prerequisite, coordinate the contradiction between life and economic interests, and when life and economic interests conflict, resolutely choose the policy most helpful to people's lives.
Other change:
- We have invited professional language editing company to carry out a new language editing of this study and re-correct some proper nouns.
- We have rewritten many unclear points in the original text. For example, the EFA method and CFA method have been used to re-check the death threat scale.
Finally, thank you again for your suggestion in our work. It is very pleasant to communicate with you on academic issues.

Reviewer 2 Report
I think there is something interesting in here but the grammar and overall command of academic English here mean it is too difficult to understand the paper in its current format and provide a useful review. The paper needs more than a proof read by a language expert, I would suggest it needs a co-author with a strong command of English to re-write this as there are many technical issues with the paper. The clearest examples of this are the phrases 'death threat' (I think you mean people facing the threat of death) and 'informational conformity consumer behavior'. I look forward to reviewing a re-written paper.
Author Response
Dear reviewer
Thanks for your review, your suggestion has a very great guiding value for our work. Follow your advice for our article, we have made major changes to the article and will report the changes to you.
Suggestion:
I think there is something interesting in here but the grammar and overall command of academic English here mean it is too difficult to understand the paper in its current format and provide a useful review. The paper needs more than a proof read by a language expert, I would suggest it needs a co-author with a strong command of English to re-write this as there are many technical issues with the paper. The clearest examples of this are the phrases 'death threat' (I think you mean people facing the threat of death) and 'informational conformity consumer behavior'. I look forward to reviewing a re-written paper.
Reply:
Thanks to the expert’s suggestion. Based on your suggestions, my co-authors and I rewrote some of the chapters in this study, reviewed and corrected some of the errors in this article, and invited a professional language editing company to edit the article.
We also used Google Scholar to query the relevant literature of death threat, and confirmed that this is a professional academic vocabulary, it is also a concept frequently used in death-related research. But at the same time, and it can also be used as the threat of death according to different usage scenarios. Related references are as follows:
- Greyson, B. (1992). Reduced death threat in near-death experiencers. Death Studies, 16(6), 523-536.
- Chambers, W. V. (1986). Inconsistencies in the theory of death threat. Death Studies, 10(2), 165-175.
Other change
- We have rewrote many unclear points in the original text. For example, the EFA method and CFA method have been used to re-check the death threat scale.
- We have added new managerial implications.
Before:
5.2 Managerial Implications
The current study provides several managerial implications for the practice of public administration in the context of the epidemic.
First, Information about the epidemic situation should be published in a timely and transparent manner, and residents should be given proper guidance. The government should give correct guidance to the products related to the epidemic in time, to help consumers understand the correct anti-epidemic products and life-related knowledge, and to avoid consumers' following the conformity consumption behavior caused by misinformation in the group. In the survey, we found that many consumers were misled by their social groups, and they hoarded many products that did not have any effect against COVID-2019. Consumers did not rationally evaluate the effects of these products but followed others and conduct the conformity consumption behavior, which not only misleads consumers on their epidemic prevention work but also causes a waste of resources. So at the beginning of the epidemic, it is vital to give consumers the correct information and the right guidance.
Second, separate the individual from the group. Individuals in the group will feel the social support given by the group, but will also get the misleading information in the group more irrationally, which will harm their epidemic prevention work. A large number of conformity consumption behaviors bring about a shortage of related products, especially in the process of epidemic prevention. A large number of people gathering will also put people at a more severe risk of infection. The government can adopt centralized management, on-demand distribution model, using logistics delivery, and other methods to deliver, to ease consumer anxiety and reduce the risk of infection of ordinary people.
After:
5.2 Managerial Implications
The current study provides several managerial implications for the practice of public administration in the context of the epidemic.
First, information about the epidemic should be published in a timely and transparent manner. In addition, the government should give correct guidance about the products related to the epidemic in a timely manner, to help consumers understand the correct anti-epidemic products and life-related knowledge, and to avoid conformity consumption behavior caused by misinformation. In the survey, we found that many consumers were misled by their social groups, and they hoarded many products that did not have any effect against COVID-19. Consumers did not rationally evaluate the effects of these products but followed others via conformity consumption behavior. This not only misleads consumers about epidemic prevention, but also causes a waste of resources. Therefore, at the beginning of an epidemic, it is vital to give consumers correct information and proper guidance.
Second, the individual should be separated from the group. Individuals in the group will feel the social support given by the group, but will also get the misleading information, which will their epidemic prevention work. A large number of conformity consumption behaviors bring about a shortage of related products, especially in the process of epidemic prevention. A large number of people gathering will also put people at more severe risk of infection. The government can adopt centralized management, on-demand distribution models, use of logistics delivery, and other methods to of delivery, to ease consumer anxiety and reduce the risk of infection.
Third, the government should intervene in the distribution of anti-epidemic products, which cannot be completely handed over to the market mechanism. In particular, the government need to use laws and regulations to crack down on hoarding, illegal trade, and maliciously driven up prices to maintain market order. The market mechanism has the characteristics of blindness and spontaneity. When the epidemic causes conformity consumer behavior, a large number of product demands will cause prices to rise sharply, and some unscrupulous merchants will use vending to conduct vicious competition. If not controlled in time, it will cause serious damage to the market order and affect consumers' confidence. Therefore, it is necessary to use legal tools to crack down on merchants using illegal methods.
Finally, when faced with the threat of death from the epidemic, people’s lives are the most important thing. When the government formulates any laws and economic management policies, it should take respect for people's lives into account as a prerequisite, coordinate the contradiction between life and economic interests, and when life and economic interests conflict, resolutely choose the policy most helpful to people's lives.
- We removed the samples under the age of 18 and re-analyzed the data. And described the research process in more detail
Before:
3.1 Survey Participants
Respondents for this research were recruited from WJX, which is the most prominent Chinese online data collection platform. The participants for this study were 1548 individuals aged 13 to 79 who participated in the consumer behavior data survey during the period of COVID-2019 in China. The data came from the participants in all 31 provinces of China undergoing COVID-2019 and had 1502 valid questionnaires. The net response rate was 97.03%. The sample included 664 female participants (44.2%) and 838 male participants (55.8%) and had a large share in income is 3000RMB to 6000RMB per month (45.87%) and a significantly higher level education (72.24% of participants had completed a college or university program). The survey also contains information on the individual, family, area characteristics, and the attitude of the government towards the virus, which we can link to the attitudes and behavior of individuals to the COVID-2019 in China.
After:
3.1 Survey Participants
Since many cities in China were subject to stay home orders due to the need for epidemic prevention at the time of the survey, this research commissioned WJX, a professional market research company, to conduct this survey. We used the random sampling from the database of WJX samples and sent questionnaire by email. The survey was conducted from February 1 to February 20, 2020. The participants for this study were1499 individuals aged 18 to 79 who participated in the consumer behavior data survey during the COVID-19 outbreak in China. The data came from the participants in all 31 provinces of China and there were 1453 valid questionnaires. The net response rate was 93.86%. The sample included 645 female participants (44.4%) and 808 male participants (55.6%) and had a large share in income is 3000RMB to 6000RMB per month (47.40%) and a significantly higher level education (73.91% of participants had completed a college or university program). The survey also contains information on individual, family, area characteristics, and the attitude of the government towards the virus(e.g. alertness level), which we can link to the attitudes and behavior of individuals.
Finally, thank you again for your suggestion in our work. It is very pleasant to communicate with you on academic issues and hope that our work will be recognized with you.
Reviewer 3 Report
This is a very interesting topic and I really enjoyed reading it. However there are a couple of places that the authors need to update, correct and revise
comments:
- line 18, research hypothesis or Hypotheses?
- line 31, nobody said toilet paper affects the COVID-19! This has another story!
- 13 to 79 years old! I think 13 years old too young for this study. Use 18+. it might also help with your AVE and Cronbach alpha.
- Death Threat is one of the Core constructs and you have used your own measurement to measure it! As you do for the other construct, you should have developed it based on theories or concepts? You have two options now: Give a more theoretical basis or do a split test. Half of the sample size to do EFA and the other half to do CFA.
- The document needs a proofreading. There are many vague statements and sentences through the manuscript, such as:
- Linke 134
- Line 298, is it a full sentence? where is the "."
Author Response
Dear reviewer
Thanks for your review, your suggestion has a very great guiding value for our work. Follow your advice for our article, we have made major changes to the article and will report the changes to you.
Suggestion 1 :
13 to 79 years old! I think 13 years old too young for this study. Use 18+. it might also help with your AVE and Cronbach alpha.
Reply:
Thanks to the expert’s suggestion. We agree with you, so in the sample we deleted all samples under the age of 18, and calculated and re-analyzed the data. The results also support the hypothesis proposed in this study.
Suggestion 2 :
Death Threat is one of the Core constructs and you have used your own measurement to measure it! As you do for the other construct, you should have developed it based on theories or concepts? You have two options now: Give a more theoretical basis or do a split test. Half of the sample size to do EFA and the other half to do CFA.
Reply:
Thanks to the expert’s suggestion. Follow your suggestion,we did a split test to confirm the reliability and validity of the death threat scale.
Before:
SPSS25.0 and AMOS 24.0 used scale structure, reliability analysis, and convergent validity, and the measurement scales employed in this research were developed and validated in a past study. For information conformity consumer behavior, we used a three-items scale adapted from Bearden et al. (1989) 's study [64], the Cronbach's α was 0.746, the AVE was 0.503,and the C.R. was 0.751. For the need to belong, we also used a three-item scale adapted from Leary et al., (2013) 's study, and in this study[65], the Cronbach's α was 0.682, the AVE was 0.643,and the C.R. was 0.843. The materialism referred to the Richins and Dawson (1992) 's study and used an eight-item scale[66], the Cronbach's α was 0.859, the AVE was 0.437,and the C.R. was 0.860. Furthermore, the perceived social support used a six-item scale referred to Canty-Mitchell and Zimet’s (2000) study [67], the Cronbach's α was 0.837, the AVE was 0.566, and the C.R. was 0.886. For the death threat, we designed a three-item scale to measure the construct. The content of this scale was that "What do you think is the risk of contracting COVID-2019?", "How scared you are of a COVID-2019" and "How much are you isolated?" the Cronbach's α was 0.733, the AVE was 0.503, and the C.R. was 0.745. All the above scale used the 5-point Likert scale, and except the need to belong (Cronbach's α=0.682, near 0.700), the Cronbach's α of other constructs all exceeded 0.700, and except for materialism (AVE=0.437, near 0.500), the AVE of other constructs all exceeded 0.500, exhibiting sufficient reliability and convergent validity (See Table 1).
After:
SPSS25.0 and AMOS 24.0 used scale structure, reliability analysis, and convergent validity, and the measurement scales employed in this research were developed and validated in a past study. For information conformity consumer behavior, we used a three-item scale adapted from Bearden et al.’s (1989) study [64]. The Cronbach's α was 0.744, the AVE was 0.504,and the C.R. was 0.745. For the need to belong, we used a three-item scale adapted from Leary et al.’s (2013) study, and in this study[65]. The Cronbach's α was 0.686, the AVE was 0.424, and the C.R. was 0.675. Materialism was measured based on Richins and Dawson’s (1992) study and used an eight-item scale[66]. The Cronbach's α was 0.860, the AVE was 0.439, and the C.R. was 0.861. Furthermore, the perceived social support was measured using a six-item scale from Canty-Mitchell and Zimet’s (2000) study [67]. The Cronbach's α was 0.839, the AVE was 0.465, and the C.R. was 0.839.
For the theat of death, according to Tobacky (1983)’s death threat scale and the situation of this study[68], we designed a three-item scale to measure the construct. The content of this scale was that "What do you think is the risk of contracting COVID-19?", "How scared you are of a COVID-19" and "How much isolated are you?" In order to verify the reliability of the death threat scale, this study randomly divided the research participants into two groups (One group has 726, and the other group has 727) and tested them with exploratory factor analysis (EFA) and confirmatory factor analysis (CFA) respectively. The results found that the EFA group’ s Cronbach’s α was 0.758, the AVE was 0.533, and the C.R. was 0.769. The CFA group’s Cronbach’s α was 0.705, the AVE was 0.482, and the C.R. was 0.724. This proves that the reliability and validity of the death threat scale are relatively stable and can be used for subsequent analysis. The Cronbach’s α of the total sample was 0.732, the AVE was 0.504, and the C.R. was 0.745.
All of the above measurements used a 5-point Likert scale, and except for the need to belong (Cronbach's α=0.682, near 0.700), the Cronbach's α of other constructs all exceeded 0.700. Except for the need to belong (AVE=0.424, near 0.500), materialism(AVE=0.437, near 0.500), and perceived social support (AVE=0.465, near 0.500), the AVE of other constructs all exceeded 0.500, exhibiting sufficient reliability and convergent validity (See Table 1).
Suggest 3 :
Some grammatical errors and wrong examples in this article.
Reply:
Thanks to the expert’s suggestion.
- We have changed the example about the toilet paper, and used the disinfectants to describe this phenomenon.
- We have invited professional language editing company to carry out a new language editing of this study and re-correct some proper nouns.
Other change:
- We have added new managerial implications.
Before:
5.2 Managerial Implications
The current study provides several managerial implications for the practice of public administration in the context of the epidemic.
First, Information about the epidemic situation should be published in a timely and transparent manner, and residents should be given proper guidance. The government should give correct guidance to the products related to the epidemic in time, to help consumers understand the correct anti-epidemic products and life-related knowledge, and to avoid consumers' following the conformity consumption behavior caused by misinformation in the group. In the survey, we found that many consumers were misled by their social groups, and they hoarded many products that did not have any effect against COVID-2019. Consumers did not rationally evaluate the effects of these products but followed others and conduct the conformity consumption behavior, which not only misleads consumers on their epidemic prevention work but also causes a waste of resources. So at the beginning of the epidemic, it is vital to give consumers the correct information and the right guidance.
Second, separate the individual from the group. Individuals in the group will feel the social support given by the group, but will also get the misleading information in the group more irrationally, which will harm their epidemic prevention work. A large number of conformity consumption behaviors bring about a shortage of related products, especially in the process of epidemic prevention. A large number of people gathering will also put people at a more severe risk of infection. The government can adopt centralized management, on-demand distribution model, using logistics delivery, and other methods to deliver, to ease consumer anxiety and reduce the risk of infection of ordinary people.
After:
5.2 Managerial Implications
The current study provides several managerial implications for the practice of public administration in the context of the epidemic.
First, information about the epidemic should be published in a timely and transparent manner. In addition, the government should give correct guidance about the products related to the epidemic in a timely manner, to help consumers understand the correct anti-epidemic products and life-related knowledge, and to avoid conformity consumption behavior caused by misinformation. In the survey, we found that many consumers were misled by their social groups, and they hoarded many products that did not have any effect against COVID-19. Consumers did not rationally evaluate the effects of these products but followed others via conformity consumption behavior. This not only misleads consumers about epidemic prevention, but also causes a waste of resources. Therefore, at the beginning of an epidemic, it is vital to give consumers correct information and proper guidance.
Second, the individual should be separated from the group. Individuals in the group will feel the social support given by the group, but will also get the misleading information, which will their epidemic prevention work. A large number of conformity consumption behaviors bring about a shortage of related products, especially in the process of epidemic prevention. A large number of people gathering will also put people at more severe risk of infection. The government can adopt centralized management, on-demand distribution models, use of logistics delivery, and other methods to of delivery, to ease consumer anxiety and reduce the risk of infection.
Third, the government should intervene in the distribution of anti-epidemic products, which cannot be completely handed over to the market mechanism. In particular, the government need to use laws and regulations to crack down on hoarding, illegal trade, and maliciously driven up prices to maintain market order. The market mechanism has the characteristics of blindness and spontaneity. When the epidemic causes conformity consumer behavior, a large number of product demands will cause prices to rise sharply, and some unscrupulous merchants will use vending to conduct vicious competition. If not controlled in time, it will cause serious damage to the market order and affect consumers' confidence. Therefore, it is necessary to use legal tools to crack down on merchants using illegal methods.
Finally, when faced with the threat of death from the epidemic, people’s lives are the most important thing. When the government formulates any laws and economic management policies, it should take respect for people's lives into account as a prerequisite, coordinate the contradiction between life and economic interests, and when life and economic interests conflict, resolutely choose the policy most helpful to people's lives.
- We have rewritten many unclear points in the original text. For example, the detailed sampling method of sample survey was added. And at the same time proofread some professional vocabulary in this study. For example, replace COVID-2019 with COVID-19.
Finally, thank you again for your suggestion in our work. It is very pleasant to communicate with you on academic issues.
Round 2
Reviewer 2 Report
Thank for your improvements however I am still recommeding further work on presentation is required. Perhaps this can be done with the journal editororial staff? The contribution, design and method are all strong.
Reviewer 3 Report
Thanks for sending back the document.
The paper is much better now. I have no further comment. Good luck with the research.